# COVID-19 pandemic-related lockdown: response time is more important than its strictness

Gil Loewenthal[1,†], Shiran Abadi[2,†], Oren Avram[1,†], Keren Halabi[2,†], Noa Ecker[1,†], Natan Nagar[1], Itay Mayrose[2,*] & Tal Pupko[1,**]

## Abstract

**The rapid spread of SARS-CoV-2 and its threat to health systems worldwide have led governments to take acute actions to enforce social distancing. Previous studies used complex epidemiological models to quantify the effect of lockdown policies on infection rates. However, these rely on prior assumptions or on official regulations. Here, we use country-specific reports of daily mobility from people cellular usage to model social distancing. Our data-driven model enabled the extraction of lockdown characteristics which were crossed with observed mortality rates to show that: (i) the time at which social distancing was initiated is highly correlated with the number of deaths, $r^2 = 0.64$, while the lockdown strictness or its duration is not as informative; (ii) a delay of 7.49 days in initiating social distancing would double the number of deaths; and (iii) the immediate response has a prolonged effect on COVID-19 death toll.**

**Keywords** coronavirus; COVID-19; epidemiology modeling; lockdown; mobility data

**Subject Categories** Computational Biology; Microbiology, Virology & Host Pathogen Interaction

## Introduction

In 2020, the coronavirus pandemic has rapidly spread around the globe, threatening health and economical systems. At first, many governments attempted to minimize exposure to the virus by limiting cross-border arrivals. However, the rapid person-to-person transmission rate of the virus (Chan *et al*, 2020; Li *et al*, 2020) required that more severe measures be taken to plummet infection frequencies. Governments that used lockdown to enforce social distancing varied in their policy, timing, and duration, in particular relative to the mortality rate in their country. For example, Italy enforced a severe, nationwide, lockdown on March 10, when over 35,000 confirmed cases and almost 3,000 deaths had already been recorded. In other countries, lockdown policies were embraced at earlier stages in attempt to prevent severe outbreaks. Israel, for instance, reached the strict lockdown on March 19 with a relatively low number of 648 confirmed cases and no deaths to that day. In contrast, several countries, such as Sweden and Japan, advocated social distancing but did not enforce a lockdown as a means of coronavirus spread prevention.

How can social distancing be quantified? One could measure governmental regulations such as the permitted walking distance from the residence, limitations on mass gatherings, school closures, and whether people were allowed to attend their workplaces. For example, Hu *et al* (preprint: 2020) suggested a score that takes into account various governmental interventions in the United States. This score was used to predict future infections depending on the intervention level. While this model may be useful when governmental decisions are made, it does not reflect whether social distancing has been implemented *de facto* (preprint: Kohanovski *et al*, 2020). Soures *et al* (preprint: 2020) used data collected from navigation applications on mobile cellphones together with past infection rates to predict future infection rates. These predictions were based on a neural-network model, in which the connection between mobility data and infection rates is hard to interpret and thus, practically, cannot be converted into tangible measures for the arms race against the disease. It is currently unknown which aspects of the lockdown (e.g., duration, strictness, timing from onset of death cases) affect mortality rates. Understanding the linkage between the lockdown dynamics and COVID-19 death incidents is highly important for balancing between health, welfare, and economy.

Location data collected from mobile phone calls have previously been linked with the identification of pandemic outbreaks, e.g., the 2005 cholera outbreak in Senegal (Finger *et al*, 2016). With the spread usage of smartphones nowadays, location and mobility data are routinely collected by numerous service providers. Mobility data from such datasets were shown to be associated with COVID-19 hotspots of disease transmission and spread (Badr *et al*, 2020; Benzell *et al*, 2020; Bonaccorsi *et al*, 2020; Kraemer *et al*, 2020; Linka *et al*, 2020; Pepe *et al*, 2020; preprint: Soures *et al*, 2020).

1 The Shmunis School of Biomedicine and Cancer Research, George S. Wise Faculty of Life Sciences, Tel Aviv University, Tel Aviv, Israel
2 School of Plant Sciences and Food Security, George S. Wise Faculty of Life Sciences, Tel Aviv University, Tel Aviv, Israel
*Corresponding author. Tel: +972 3 640 7212; Fax: +972 3 640 9850; E-mail: itaymay@post.tau.ac.il
**Corresponding author. Tel: +972 3 640 7693; Fax: +972 3 640 9245; E-mail: talp@tauex.tau.ac.il
†These authors contributed equally to this work

Here, we develop parametric models that quantify trends related to mobility and mortality and fit them to all OECD countries. Using these models, we demonstrate that the correlation between the timing in which the social distancing was initiated and the COVID-19-related deaths is $r^2 = 0.64$ across the OECD countries excluding Japan (that was previously reported as an exception with respect to the spread of the disease, e.g., by Iwasaki & Grubaugh, 2020). In contrast, the severity of the lockdown and its duration are not as informative for explaining mortality rates. Our analysis thus suggests that a moderate lockdown, rather than a very strict one as was imposed by most countries, should be sufficient to curb COVID-19-related mortality, as long as action is taken in the appropriate time frame.

# Results

Following the COVID-19 outbreak, Apple Inc. has started publishing daily reports regarding people mobility, collected from usage of maps on mobile cellphones (Data ref: Apple, 2020). We used these mobility data, denoted as $M(t)$, to quantify the actual commencement of the lockdown as a function of time in different OECD countries. We collected daily death incidents across time and overlaid them on the mobility data (see Fig 1A for the United Kingdom as a representative OECD country and Appendix Fig S1 for all OECD countries). We observed that the trend of daily deaths stabilized and subsequently decreased several days after a sharp mobility drop, typically observed in March, corresponding to the time of applying governmental interventions. During the time period between January and May, most countries enforced social distancing as a strategy to handle the initial outbreak. Following this period, with the accumulation of additional knowledge regarding means of prevention and treatment (Sanders et al, 2020; Xu et al, 2020) and as many countries started to relax the restrictions and ease the lockdown, the mobility trends across countries have diverged. For example, the mobility trend in Israel returned to the baseline and did not dramatically fluctuate after May, while in Sweden it rose beyond the baseline and declined back toward August (see Appendix Fig S1 for the trends of the OECD countries between January and August).

## Mobility analysis

To model the social distancing dynamics during the initial phase of the pandemic, we focused on the time period between January 13 and May 10 (termed the "lockdown period" hereafter). Inspection of the mobility trends during this time period revealed four phases: (i) a stable phase of high mobility (with fluctuations on weekends); (ii) a sharp drop (suggesting social distancing has actually started); (iii) a period of low mobility; and (iv) a gradual incline toward a normal routine (Fig 1). Phases (i)-(iii) resemble a (mirrored) logistic function and phase (iv) is approximately linear. We modeled this overall trend by assembling a logistic function and a linear one as a function of time ($t$, given in days):

$$\widehat{M}(t) = \begin{cases} \frac{L}{1+e^{-k(t-t^0)}} + b & t \leq t^1 \\ a(t - t^1) + \widehat{M}(t^1) & t > t^1 \end{cases}$$

The six free parameters of this model are illustrated in Fig 1B.

Fitting the mobility model to the 37 OECD countries resulted in an average $r^2$ of 0.9 between the observed data and the fitted functions (all $P$ values < $10^{-32}$, Appendix Table S1). The inferred model parameters enabled the comparison of several informative features for the different countries (see Materials and Methods). As examples, we present the fitted models for five representative OECD countries: Germany, Israel, Italy, Spain, and Sweden (Fig 2; for the inferred features and fitted models of all countries see Appendix Table S2 and Appendix Fig S2). Our results demonstrate that while the *lockdown strictness* varied considerably, all countries reached some form of a lockdown by the middle of March 2020, with Spain presenting the most intense drop (88%). The *social distancing start time* in Italy occurred earlier, on February 25 compared with March 6–9 for the abovementioned four other countries. Nevertheless, the mobility in Italy declined in a relatively gradual manner with respect to other examined countries, as the *drop duration* lasted 20 days. The extent of mobility reduction in Germany (59%) was relatively low compared to other countries in which a lockdown was issued, and a gradual return to normal routine was initiated right after the lowest mobility level was reached. Even though a lockdown was not regulated in Sweden, the data and model demonstrate that social distancing indeed happened, as a drop of 29% was observed followed by a moderate return back to routine (*lockdown release rate* of 0.57).

## COVID-19 mortality

We examined the effect of the extracted mobility features on the dynamics of the mortality levels during the lockdown period. We focused on the lockdown period to examine the effect of the lockdown as the main measure, without the effect of other obscuring means of prevention that were learned and adopted after the lockdown was eased. Notably, toward the end of the lockdown period, different countries were at different phases of the daily mortality trends. For example, Greece and Australia reached only few daily new death cases, while in Germany and Italy the decline was more gradual and in Mexico and Columbia the trends were still elevating (Appendix Fig S1). To compute the expected mortality rate across time, we fitted a logistic function, denoted as $\widehat{D}(t)$, to the accumulated number of COVID-19 deaths of each country across time, $D(t)$:

$$\widehat{D}(t) = \frac{L_d}{1 + e^{-k_d(t-t_d^0)}}$$

as in Tátrai and Várallyay (preprint: 2020). The parameters $L_d$, $k_d$, and $t_d^0$ are similar to those defined for the mobility model and represent the total expected mortality at the end of the pandemic, the mortality increase rate, and the time the cumulative mortality has reached its midpoint, respectively. This enabled to compute the *COVID-19 Mortality Probability*, namely, the expected mortality normalized by the population size of each country. The fitting of $\widehat{D}(t)$ to $D(t)$ across countries resulted in an average $r^2$ of 0.99 (max $P$ value = 1e-96; Appendix Table S3; see Fig 3 for examples of Israel and Japan and Appendix Fig S2 for all countries).

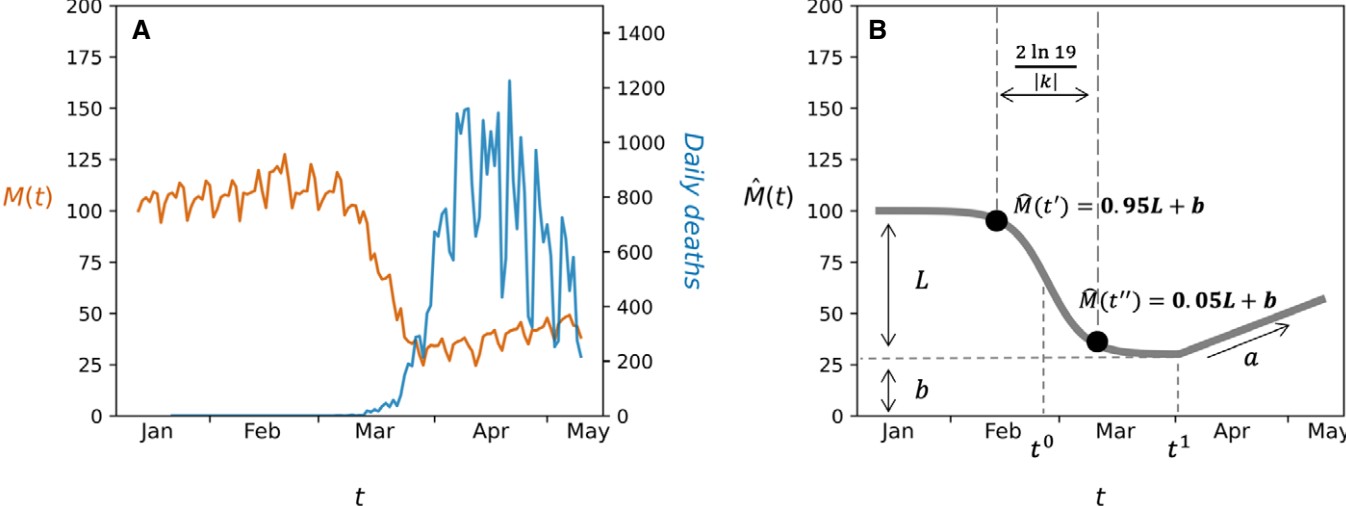

**Figure 1. Modeling mobility data.**

A   Daily mobility data, $M(t)$, (orange line, left $y$-axis) overlaid with daily deaths (blue line, right $y$-axis) for the United Kingdom during the lockdown period (January 13 to May 10). $M(t)$ is given as percentages relative to that recorded on January 13, which serves as the baseline. For the data of all OECD countries until August 31, see Appendix Fig S1.

B   An illustration of the mobility model and its free parameters: $L$—mobility difference between routine and lockdown; $k$—drop steepness; $t^0$—drop midpoint; $b$—mobility level during lockdown; $t^1$—release day; $a$—recovery rate. $t'$ represents the *Social distancing start time*, and $t''$ represents the *Minimal mobility time point*, corresponding to the times before and after the mobility drop, respectively.

## Association between mobility and mortality data

We computed the response time of each country, $\tau$, defined as the difference between the *social distancing start time* and the day in which ten first deaths were recorded. Fig 3 demonstrates the computation of $\tau$ for Israel and Japan ($\tau = -19.83$ and 18.16 days, respectively; see Appendix Fig S2 and Appendix Table S4 for all countries). While a negative $\tau$ was inferred for most countries, a positive $\tau$ was inferred for five countries (France, Italy, Japan, Spain, and the United States), indicating that social distancing started after ten COVID-19 deaths were documented (Fig 4, Appendix Fig S2). We observed a significant correlation between $\tau$ and the log *COVID-19 Mortality Probability* ($r^2 = 0.38$, $P$ value = 1e-4). Previous reports have discussed the abnormally low mortality rate in Japan (Iwasaki & Grubaugh, 2020); thus, we computed the correlation excluding Japan and obtained a substantial increase in correlation ($r^2 = 0.64$, $P$ value = 1e-8). Neither the *lockdown strictness* nor the *lockdown duration* was significantly correlated with log *COVID-19 Mortality Probability* (Table EV1).

The high correlation between $\tau$ and the log *COVID-19 Mortality Probability* yielded a crucial implication, as it allowed inferring the time required for this probability to double. We fitted a linear regression to the data presented in Fig 4 (excluding Japan) and used the slope of the fitted regression line to compute the estimated time for doubling the *COVID-19 Mortality Probability*. Accordingly, our results indicate that a 7.49 days delay in lockdown commencement doubled the expected number of deaths (95% CI [6.02, 10.03]). This result, which emerged from a data-driven model, is in accordance with the results of an epidemiological-model based study (preprint: Pei *et al*, 2020), which concluded that 54% of the deaths in the

United States could have been prevented if non-pharmaceutical interventions had been implemented a week earlier.

We focused our analysis on 37 OECD countries, to concentrate on a representative group of relatively reliable reports. Nevertheless, our results sustain when including additional non-OECD countries or when concentrating on subregions for which sufficient data exist: the $r^2$ between $\tau$ and the log *COVID-19 Mortality Probability* for 58 countries was 0.37 ($P$ value = 4e-7; Fig EV1). A significant correlation was also observed when analyzing states within the United States ($r^2 = 0.36$; $P$ value = 8e-6; Fig EV2). We next examined whether our conclusions hold when the infection rate, rather than the mortality rate, is examined. To this end, we fitted $\hat{D}(t)$ to the accumulated number of COVID-19 confirmed cases across time and computed the log *COVID-19 Infection Probability*, similar to the way the log *COVID-19 mortality Probability* was computed. A significant correlation of $r^2 = 0.47$ was also observed between $\tau$ and the log *COVID-19 Infection Probability* in OECD countries ($P$ value = 4e-6; Fig EV3, Table EV2). Notably, the infection rate is highly dependent on the COVID-19 test policy and thus varies across countries.

## Prolonged impact of the initial response on the COVID-19-related mortality

Evidently, the presented analysis corresponds to the lockdown taken as an initial response by most countries in the first several months of the pandemic. Next, we examined whether the effect of the initial response sustained over a prolonged time period. To this end, we extracted the reported mortality rates on August 31, 2020, and normalized them by the population size (termed, Aug-20 *COVID-19 Mortality Probability*). A significant correlation between $\tau$, as

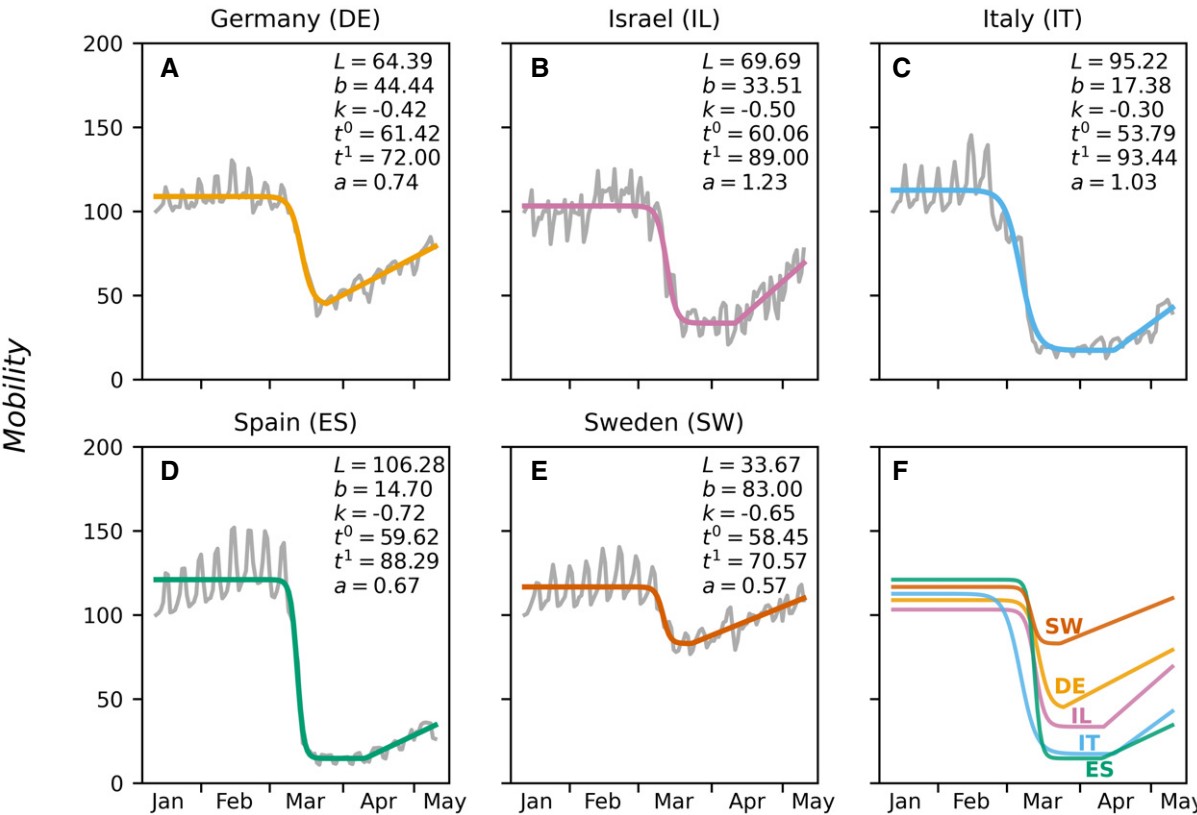

**Figure 2. The fit of the mobility model for five representative OECD countries.**

A–F Colored lines in panels (A-E) represent the mobility model $\hat{M}(t)$ fitted to the mobility data $M(t)$ (gray lines). The optimized parameters are indicated. Panel (F) presents the overlay of the five fitted models. The two-letter codes and the five colors correspond to the countries represented in panels (A-E) (countries abbreviations are denoted in the titles of the panels). The x-axes represent days from January 13 to May 10, 2020. The y-axes represent the percentage change in mobility. For the parameter values and the inferred features of all 37 countries, see Appendix Tables S1 and S2.

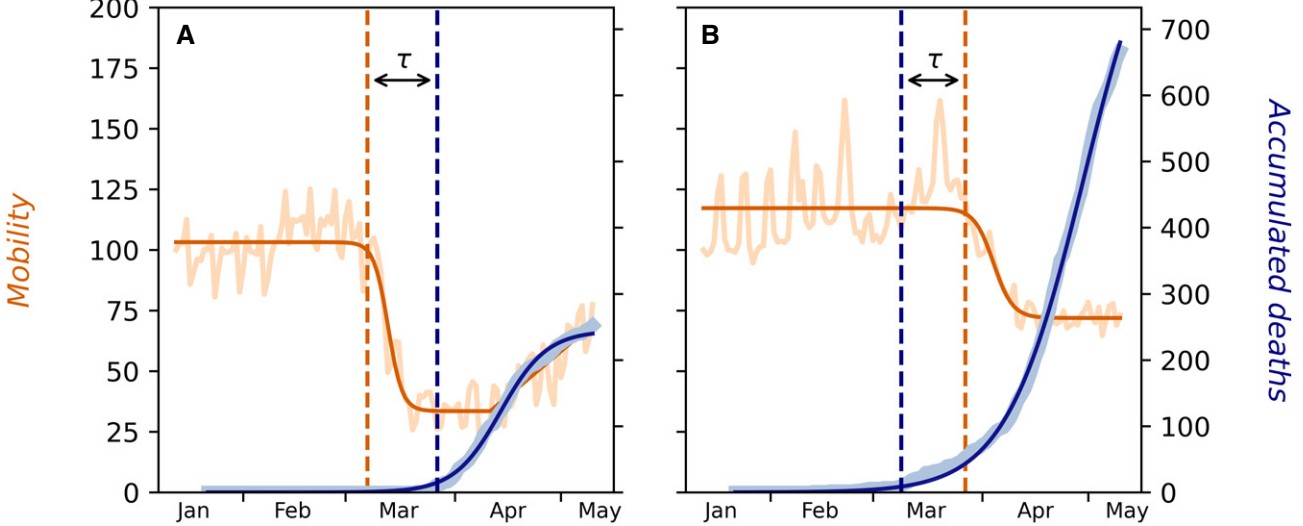

**Figure 3. Synchronizing between the mortality model and the mobility model.**

A, B The dark orange plots represent the mobility model, $\hat{M}(t)$, fitted to the mobility data, $M(t)$ (light orange; left y-axis) of (A) Israel and (B) Japan. The dashed vertical orange line represents the *social distancing start time*, predicted by the mobility model. The dark blue plots represent the mortality model, $\hat{D}(t)$, fitted to the accumulated death data, $D(t)$ (light blue; right y-axis). The dashed vertical blue lines represent the day ten deaths were documented. $\tau$ represents the time difference between the orange and the blue vertical lines, defined as the response time ($\tau$ is negative for Israel and positive for Japan). The graphs for all OECD countries are given in Appendix Fig S2.

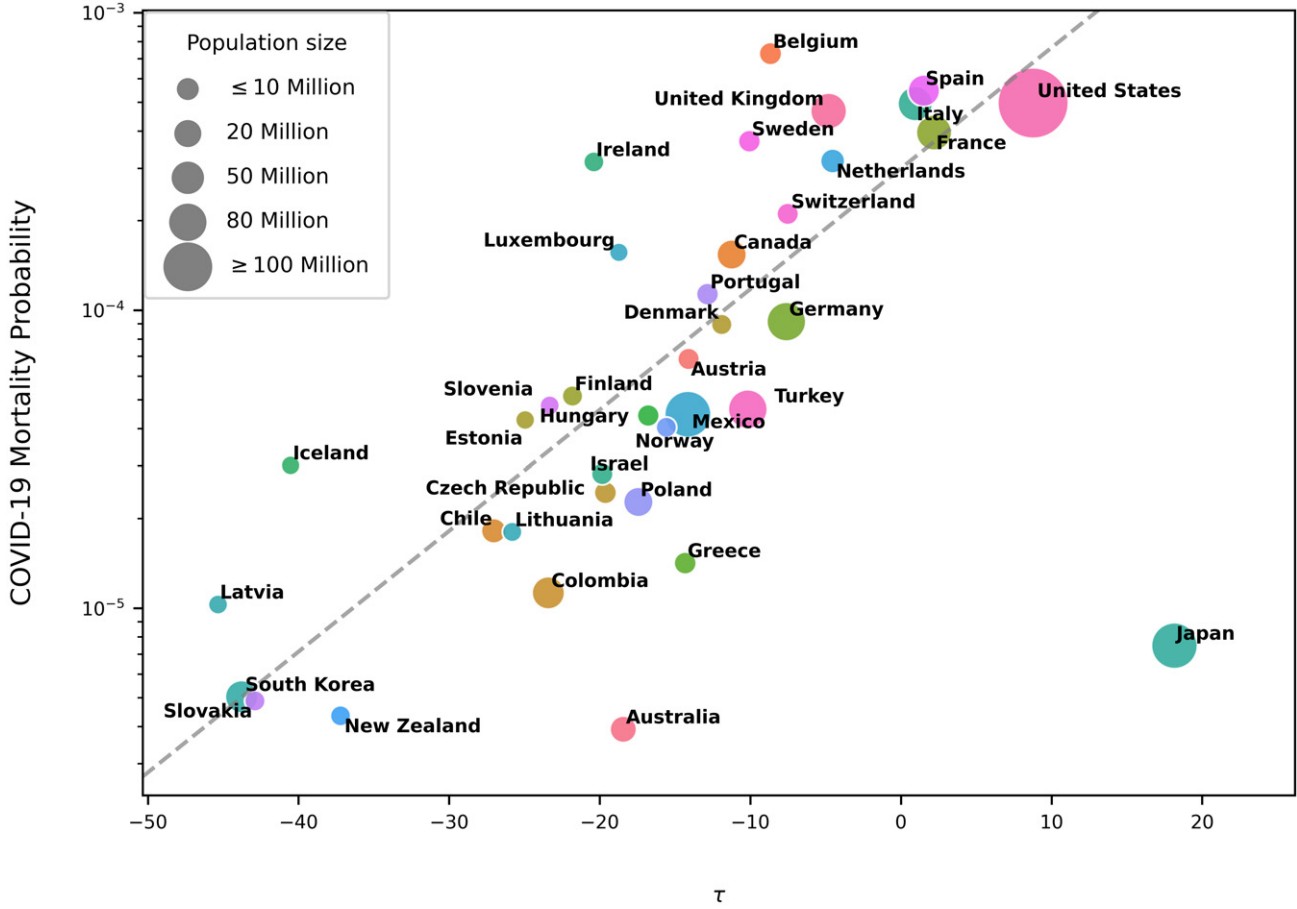

**Figure 4. A semi-logarithmic scatter plot of the *COVID-19 Mortality Probability* and τ.**

The x-axis represents τ, the difference between the *social distancing start time* and the day in which the ten first deaths were recorded for the respective country (intuitively, the response time). The y-axis represents the *COVID-19 Mortality Probability* in a logarithmic scale. Dot sizes are proportional to population sizes. The dashed line corresponds to the fitted regression, excluding Japan. For raw data, see Appendix Table S4.

computed from fitting $\widehat{M}(t)$ to the mobility data during the lockdown period, and the log Aug-20 *COVID-19 Mortality Probability*, was maintained ($r^2 = 0.62$, *P* value = 1e-8 and $r^2 = 0.34$, *P* value = 2e-4 when excluding and including Japan, respectively; Fig EV4). Still, neither the *lockdown strictness* nor the *lockdown duration* was significantly correlated with the log Aug-20 *COVID-19 Mortality Probability* (Table EV3). The significant correlation sustained when the log Aug-20 *COVID-19 Mortality Probability* was examined across the 58 countries for which data are available, across the United States countries, and when the log Aug-20 *COVID-19 Infection Probability* was examined (Appendix Figs S3–S5). Altogether, these analyses imply that the initial response was critical to curb total COVID-19-related mortality and had a long-term impact.

## Discussion

In this study, we modeled the mobility dynamics across time during the COVID-19 pandemic. Using this model, we computed explanatory features that characterize a lockdown, and in turn, these features provided a quantitative measure for comparing the

lockdown dynamics and outcome across countries. We found high correlation between the response time of a country and its mortality rate. This finding suggests that countries that took early measures to limit population mixing had better control on the viral-related mortality. While these conclusions were derived for the lockdown period, i.e., in the midst of the pandemic when the mortality rates could roughly be predicted, accumulation of more recent data demonstrates that the initial lockdown response time has a prolonged impact on mortality rates. In contrast, neither the *lockdown duration* nor the *lockdown strictness* was significantly correlated with the mortality rates (Tables EV1 and EV3). These results imply that a tight lockdown has been unnecessary and that the immediate response was of utmost importance.

Mobility data collected from location identification of various smartphone applications have been previously analyzed in relation with the COVID-19 pandemic, e.g., to better understand the importance of travel restrictions on the infection rate or to construct platforms for capturing movements between provinces for decision making (Badr *et al*, 2020; Benzell *et al*, 2020; Bonaccorsi *et al*, 2020; Kraemer *et al*, 2020; Linka *et al*, 2020; Pepe *et al*, 2020; preprint: Soures *et al*, 2020). All of these studies proved that

collecting mobility data is of high importance for controlling the trajectories of the pandemic. However, they focused on limited geographical areas and none has modeled the data through time in order to extract meaningful features of the lockdown. Official governmental regulations are possible resources for extracting features regarding the lockdown. Such features were examined for their association with infection rates in previous studies (Flaxman et al, 2020; Gatto et al, 2020; preprint: Hu et al, 2020; Li et al, 2020). However, the time governmental regulations were declared was often significantly different from the date social distancing initiated de facto (preprint: Kohanovski et al, 2020). For example, Italy announced regional lockdowns in three phases. Initially, a lockdown was applied in Lombardy and Veneto regions on February 20. This lockdown was expanded to all of Northern Italy on March 8, and finally to a nationwide lockdown on March 11. The mobility data show that even though the initial lockdown was declared on Lombardy, mobility volumes across Italy kept elevating and started to decrease only a week later. In contrast, when the lockdowns on Northern Italy and later, on the entire country, were declared, a large drop in mobility usage had already taken place (Appendix Fig S6).

Notably, the analysis conducted in this study corresponds to the initial period of the COVID-19 pandemic, when a small portion of the community was immune to the disease. The mortality model suggested here, $\widehat{D}(t)$ (similar to preprint: Tátrai & Várallyay, 2020), fits the dynamics observed during this period. With the accumulating information regarding means of prevention and treatment, as well as with the growing proportion of immunity within the population, the trends have changed within and across countries (Linka et al, 2020). Therefore, we do not expect that the mortality model would fit more recent and future mortality data. However, by extracting the reported death tolls approximately 4 months after the lockdown period (August 31, 2020), we validated that our conclusions still hold (Fig EV4). That is, our analyses suggest that the immediate lockdown response time to the pandemic outbreak is highly correlated to the death tolls in the long run.

Most of the examined OECD countries complied well with the regression analysis in this study. Small deviations could be explained by modest variations between countries, such as the conditions for defining a patient as a SARS-CoV-2 carrier, or by differences in mobile usage across areas. Notably, different geographical areas also differ in numerous other attributes that may affect the coronavirus spread and induced mortality rates, e.g., humidity, wind speed, ethnicity, viral genotypic variation, and cultural habits (Coccia, 2020; Jüni et al, 2020). One eminent example emerging from our analysis is Japan, where a relatively low mortality rate occurred even though mobility reduction took place relatively late. The low mortality rate in Japan has previously been discussed (Iwasaki & Grubaugh, 2020), and it should be beneficial to better understand the different trajectory of the pandemic in Japan, with respect to the Japanese governmental regulations and customs as a possible alternative to a strict lockdown. Combining these features with the proposed mobility model may increase its overall accuracy.

The results of our analysis show that a delay of 7.49 days in initiating social distancing would lead to doubling the total expected number of deaths. This finding resembles previous results regarding the pandemic doubling time, which is described as the time that passes until the number of confirmed cases at a given time point is doubled (Kraemer et al, 2020; Li et al, 2020; Wu et al, 2020). The

following theoretical scenario may explain the similarity between the two results: if a lockdown is initiated and assuming that it almost completely abolishes new infections, the number of infections remains unchanged and so is the expected mortality. If a lockdown is not initiated, we expect that the number of infections would be doubled after a time period that is equal to the doubling time. Assuming a fixed percentage of death cases, we also expect a doubling of the total mortality. Therefore, under these two assumptions, initiating a lockdown as early as a period equal to the doubling time would result in half as many death incidents. Other studies estimated the doubling time to be lower than 4 days (Lurie et al, 2020; Muniz-Rodriguez et al, 2020; Silverman et al, 2020). Notably, all of the aforementioned studies were conducted on infection or mortality data collected during different time phases between December 2019 and the beginning of March 2020, and it is possible that the effective doubling time diverged across countries and through time.

The results of our analysis show that social distancing is a major factor in controlling COVID-19 spread. However, it also shows that a strict lockdown policy is not required. Therefore, to avoid major infection outbreaks, we suggest undertaking a moderate form of a lockdown that can be tolerated by the society for longer time periods, with minimal socioeconomic damage.

# Materials and Methods

### Mobility data

Mobility data, $M(t)$, with one data point per day, $t$, were downloaded from Apple repository on August 31, 2020 (Data ref: Apple, 2020). The Apple dataset reports the daily volume of directions requested from Apple maps on mobile cellphones for driving, walking, or using transit (public transportation) in a specified region. The amount of requests per day is reported as the percentage with respect to a benchmark (100%) set on January 13, 2020. For extracting features that characterize the lockdown, we focused our analysis on the lockdown period (January 13 to May 10, corresponding to 119 data points). Due to the high similarity between "walking" and "driving" data during the lockdown period (average correlation across countries $r^2 = 0.91$, SD = 0.07, max $P$ value = $10^{-129}$) and since the "transit" data are incomplete, all analyses were applied using the "driving" data only.

To fit $\widehat{M}(t)$ to the mobility data $M(t)$ during the lockdown period and to infer the values of the parameters for every country, we used the Levenberg–Marquardt optimization algorithm from the SciPy module (Levenberg, 1944; Marquardt, 1963; Virtanen et al, 2020). According to the parameters inferred from $\widehat{M}(t)$, we computed seven features to characterize the mobility trend in a country, as follows: (i) $t'$, Social distancing start time; and (ii) $t''$, Minimal mobility time point, corresponding to the times before and after the mobility drop. These points are defined as time 95% and 5% of the drop, parameterized by $L$, and $t^0$ is the middle time point between them (see Fig 1A). Thus, $\widehat{M}(t') = 0.95L + b$ and $\widehat{M}(t'') = 0.05L + b$ Then,

$$\widehat{M}(t') = \frac{L}{1 + e^{-k(t-t^0)}} + b = 0.95L + b$$

$$-k(t' - t^0) = \ln\frac{0.05}{0.95} = -\ln 19$$

$$t' = t^0 + \frac{\ln 19}{k}$$

Similarly, $t'' = t^0 + \frac{\ln 19}{k}$ (note that $k$ is negative); (iii) *Drop duration*, the time difference $t'' - t' = \frac{2\ln 19}{k}$; (iv) *Lockdown release day* $t^1$; (v) *Lockdown strictness* $\frac{L}{L+\hat{M}(t^1)} \times 100$, such that $\hat{M}(t^1)$ is the function value at the release day; (vi) *Lockdown duration* $t^1 - t''$; (vii) *Lockdown release rate a* (the slope of the linear function).

### Mortality data

The daily cumulative numbers of COVID-19-related mortalities, $D(t)$, were downloaded from the COVID-19 Data Repository by the Center for Systems Science and Engineering (CSSE) at Johns Hopkins University (Dong *et al*, 2020). The data were available for the period of January 22 to August 31, 2020 (a total of 223 data points). The cumulative number of deaths in Australia, Canada, and the United States were aggregated across the regions reported in the dataset within each of these countries. To compute the expected mortality, we fitted a logistic function $\hat{D}(t) = \frac{L_d}{1+e^{-k_d(t-t_d^0)}}$ to $D(t)$ limited to the lockdown period (until May 10, corresponding to 110 data points), as in Tátrai and Várallyay (preprint: 2020), using the Levenberg–Marquardt optimization algorithm from the SciPy module (Levenberg, 1944; Marquardt, 1963; Virtanen *et al*, 2020). The parameters $L_d$, $k_d$, and $t_d^0$ are similar to those defined for the mobility model and represent the total expected mortality at the end of the pandemic (as a prospective relative to the lockdown period), mortality increase rate, and the time the cumulative mortality has reached its midpoint, respectively. To infer the *COVID-19 Mortality Probability*, the expected mortality ($L_d$) was normalized by the population size for each country. Data of population size were obtained from the World Population Review website (Data ref: World Population Review, 2020). Of note, our goal in this work was not to predict mortality rates, but rather, to find correlates with large changes in mortality patterns across countries. Since we correlate with the logarithm of the mortality rates, we expect that small deviations in mortality estimates will not affect our conclusions.

For a similar analysis of the infections probability (Fig EV3, Table EV2), the reported daily confirmed cases were downloaded from the COVID-19 Data Repository by the Center for Systems Science and Engineering (CSSE) at Johns Hopkins University (Dong *et al*, 2020) and were fitted to $\hat{D}(t)$ in a similar procedure.

For the analysis of the prolonged impact of the initial response on the COVID-19-related mortality, the raw mortality and infection rates reported on August 31, 2020, were normalized by the population size for each country.

### Association between mortality and response time

We define a response time, $\tau$, as the difference between two time points: the *social distancing start time* (as inferred from $\hat{M}(t)$) and the day in which ten first COVID-19-related deaths were recorded (according to $D(t)$, see Fig 3). The ten deaths threshold was set to avoid incidental fluctuations that do not reflect the mortality trend of a certain country (see Fig EV5 for the results using different thresholds). We chose an absolute threshold rather than a relative threshold (i.e., number of death incidents normalized to population size) because the very initial dynamics of the disease are not expected to

### The paper explained

**Problem**

In order to curb the spread of the COVID-19 pandemic, governments around the world have enforced mobility restrictions on their citizens. These mobility restrictions included, for example, closure of non-essential businesses and prevention of public gatherings and led to serious socioeconomic consequences. We wished to understand the impact of mobility restriction on mortality rate, by comparing mobility and mortality data across countries around the world.

**Results**

We analyzed mobility volume obtained from cellular usage of Apple users from many countries around the world to quantify country-specific lockdown characteristics, such as, social distancing start time, lockdown timing, lockdown strictness, lockdown duration, and lockdown release rate. We crossed the different characteristics with the observed mortality rate of each country. Our analysis suggests that the time at which social distancing was initiated had a critical and long-term effect: a delay of 7.49 days in lockdown commencement is associated with a doubling of the expected number of deaths. This is in contrast to other parameters such as the lockdown strictness that had negligible impact on mortality.

**Impact**

Countries that enforced a very strict lockdown could have obtained similar mortality figures with less stringent mobility restrictions as long as social distancing is initiated as early as possible after the first incidents are recorded. As a direct consequence, the socioeconomic damage of a strict lockdown could have been less severe.

be strongly coupled with the population size. Furthermore, setting a relative threshold of one death per 1 million (or more) citizens is problematic for countries such as Iceland because its population size is smaller than $10^6$. However, setting a threshold of one death per $10^5$ citizens is problematic for countries with relatively low number of deaths because such countries approached the starting threshold relatively late (i.e., the mortality rate in Australia was $3 \times 10^{-5}$ death cases per population size on August 31, 2020).

We fitted a linear regression model to these data (Fig 4; excluding Japan), i.e., between $\tau$ and the log *COVID-19 Mortality Probability* (denoted as $f(\tau)$). This fitting resulted in the inferred model: $\log_{10} f(\tau) = 0.04\tau - 3.52$ (slope 95% CI [0.03, 0.05]). Let $\tau'$ be an arbitrary response time point and let $\tau''$ be the time with twice the number of deaths, i.e., $f(\tau'') = 2f(\tau')$. Therefore,

$$\frac{f(\tau'')}{f(\tau')} = \frac{10^{0.04t''-3.52}}{10^{0.04\tau'-3.52}} = 10^{0.04(\tau''-\tau')} = 2$$

$$\tau'' - \tau' = \frac{\log_{10}2}{0.04} = 7.49$$

Resulting in a doubling time of 7.49 days with 95% CI [6.02, 10.03].

## Data availability

The code developed in this study is available at: https://github.com/shiranab/COVID-19-Mobility-analysis.

**Expanded View** for this article is available online.

## Acknowledgements

GL, SA, OA, KH, and NN were supported in part by a fellowship from the Edmond J. Safra Center for Bioinformatics at Tel Aviv University. SA was partly supported by the Rothschild Caesarea Foundation. OA was partly supported by the Dalia and Eli Hurvits Foundation. The authors wish to thank Itsik Pe'er for comments.

## Author contributions

GL and OA conceived the project; GL, SA, OA, KH, NE, and NN examined possible data sources and performed the analysis; SA wrote the primary draft; GL, SA, and OA completed the manuscript writing with the contributions from all authors; IM and TP supervised the study.

## Conflict of interest

The authors declare that they have no conflict of interest.

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
