## [Review Process File · EMBO Molecular Medicine]

COVID-19 pandemic-related lockdown: response time is more important than its strictness

Gil Loewenthal, Shiran Abadi, Oren Avram, Keren Halabi, Noa Ecker, Natan Nagar, Itay Mayrose, and Tal Pupko

DOI: [10.15252/emmm.202013171](https://doi.org/10.15252/emmm.202013171)

Corresponding author(s): Tal Pupko (talp@tauex.tau.ac.il) , Tal Pupko (talp@tauex.tau.ac.il)

Review Timeline:

Submission Date:	23rd Jul 20
Editorial Decision:	20th Aug 20
Revision Received:	6th Sep 20
Editorial Decision:	16th Sep 20
Revision Received:	21st Sep 20
Accepted:	23rd Sep 20

Handling Editors: Zeljko Durdevic / Celine Carret

Transaction Report:

20th Aug 2020

Dear Dr. Avram,

Thank you for the submission of your manuscript to EMBO Molecular Medicine. We have now heard back from the two referees whom we asked to evaluate your manuscript. Although the referees find the study to be of interest, they also raise a number of questions and concerns that need to be addressed in a revised article.

You will see that referee 1 recommends updating the findings with more recent data, and we agree that given the fast evolving situation across the globe, it would be really interesting to have an updated image of the findings. Referee 2 recommends providing more details and explanations as requested, and updating the literature. Together with referee 1 upon cross-commenting, we agree that this would strengthen the article and possibly increase its impact.

However, while ref. 2 recommends removing the time between infection to fatality part of the analysis as it doesn't appear to add too much to the results, ref.1 suggests instead to provide "a good explanation (and possible re-analysis) as it is likely that there would be confounding factors that may distort the number and, which, ideally would be corrected for, but at least identified".

We would therefore welcome the submission of a revised version within three to six months for further consideration and would like to encourage you to address all the criticisms raised as suggested to improve conclusiveness and clarity. Please note that EMBO Molecular Medicine strongly supports a single round of revision and that, as acceptance or rejection of the manuscript will depend on another round of review, your responses should be as complete as possible.

***** Reviewer's comments *****

Referee #1 (Comments on Novelty/Model System for Author):

Methodology, models deployed and analyses made are sensible and appropriate. Results are interpreted well and within the boundaries of the experimental settings

Referee #1 (Remarks for Author):

This is a timely, relevant, methodologically sound and well carried out study. Given the time gap between the work done and potential publication, authors would be encouraged to update, in as much as possible [as this is, of course, a developing event], their results considering the data generated since completion of the work. Also because several countries have shown new surges since then. Furthermore, substantial data from non-OECD countries have become available since then. An updated analysis (which would expand and either consolidate or enable to refine the

conclusions, would definitely increase the value and timeliness of the work.

Referee #2 (Comments on Novelty/Model System for Author):

While this has been studied a lot, the authors have done a better job than any other manuscript on the topic I have seen. The importance of this topic is pretty much self-evident.

Referee #2 (Remarks for Author):

In their manuscript, Loewenthal and co-authors introduce an elegantly simple mathematical approach for exploring the relationship between COVID-19 deaths and social distancing (measured by proxy cellphone mobility indices). This article is a nice contribution but there are a things the authors still need to address. Overall, the conclusions of the paper that I believe the authors have shown are directly tied to Figure 4. That is, I believe that the authors have shown that time to social distancing (as measured by cellphone mobility) is related to mortality (That a great conclusion!). However, I don't buy some of the other stated conclusions in the paper.

I don't buy the authors claims about the time between infection to fatality. The authors try to make a claim about this lag based on a proxy (the offset between mobility reduction and the decline in mortality growth rate). This is too much of a stretch for me, there are just too many sources of potential confounding that make this proxy not a good measure of the lag between infection to fatality. To underscore my point, it seems ridiculous to me that there would be so much variation from country to country in this value (e.g., 38-39 days in Finland, Hungary, and Estonia) but only 8 in the United Kingdom? I would recommend authors remove this section and its claims to strengthen the paper. I don't think the paper needs this added conclusion to be impactful.

The abstract mention that the time at which social distancing was initiated explains 62% of the variation of the number of deaths. Where are the authors pulling this statistic from, it was not clear to me in the manuscript where this came from? I am skeptical of this statistic.

Regarding the 7.49 day delay causing a doubling in deaths: Are the authors implying this statistic (7.49 days) is related to the doubling time of the epidemic? It would seem like this based on their citation of the 7.4 day from Li et al. The 7.4 day doubling time is not a great estimate. That 7.4 day estimate is not a good estimate

(<https://www.nytimes.com/interactive/2020/03/21/upshot/coronavirus-deaths-by-country.html>; Silverman et al. Science Translational Medicine 2020 for the US at least). Initial doubling times were closer to every 3 days. However, it seems to me that what the authors are really measuring is not the same doubling time as mentioned in Li et al as it is more a function of the effective doubling time around the time that the interventions were occurring (not the initial epidemic doubling time).

What does the Apple mobility data actually measure? "movement" is not directly measured by cell phones. That is, is it based on a pedometer in the phone? Is it based on the maximum diameter of the convex hull of the cell phone's position on a given day? As far as I know Apple does not make this clear and it is a crucial limitation of this data -- What is the data?

I applaud the authors for not just assuming exponential growth dynamics (e.g., SIR models etc...) in their manuscript. However, the authors should still describe how a depletion of susceptibles could alternatively explain some of the death dynamics observed in this manuscript. For example, the growth rate of deaths must (on average) decrease as the susceptible population is exposed to the

virus. As it currently stands, it seems the authors are assuming that all changes in deaths could be attributable to social distancing without accounting for the fact that there are other processes that can dictate the dynamics of deaths. I think what the authors have done is probably ok for the time-window they study but I don't think this implicit assumption would hold later in the epidemic when the resistant population is non-trivially large, this should be explicitly discussed.

Regarding the authors literature review: The relation between social distancing / intervention and deaths / cases is probably one of the most studied parts of COVID-19. In fact, to the best of my knowledge, numerous authors have looked at the relation between mobility data (as a proxy for social distancing) and cases/deaths. The authors only cite Soures et al. A brief google search brings up a bunch of relevant other citations (https://scholar.google.com/scholar?hl=en&as_sdt=0%2C39&q=mobility+covid-19&btnG=).

Minor Comments:

Throughout the manuscript the authors need to be a little more precise with their discussion of R^2 statistics. The square of the pearson product moment (between observed and fitted) would be an R^2 statistic. The authors just need to tighten up the language on this point as it is confusing the way they have written it.

The authors only describe the definition of COVID-19 Mortality Probability in the methods. This is a key statistic in their paper and needs to be made clear in the results.

On page 5, the authors must specify that their statement that the phases reflect a certain functional form is a subjective determination. E.g., it looks logistic to us. I am fine with this as I like the functional form they picked, but still, it is subjective and that must be mentioned. The current wording makes it seem objective.

Page 6, the authors should show the functional form of the logistic function for COVID-19 mortality. "See Methods" is fine if used sparingly but the authors do it too much and it makes the paper hard to read.

Page 9, is σ the variance of the estimate? This notation is not defined.

The new results the authors introduce in the discussion (last few paragraphs) should be in the results. The discussion should not introduce new data but summaries the data.

The authors should denote t' and t'' in Figure 1 as it is confusing with the concomitant notation for t_0 and t_1 .

Page 13, The results of changing the threshold of deaths (from 10 to 5 or 20) should include a reference to the relevant figures in the supplement.

Referee #1 (Remarks for Author):

This is a timely, relevant, methodologically sound and well carried out study. Given the time gap between the work done and potential publication, authors would be encouraged to update, in as much as possible [as this is, of course, a developing event], their results considering the data generated since completion of the work. Also because several countries have shown new surges since then. Furthermore, substantial data from non-OECD countries have become available since then. An updated analysis (which would expand and either consolidate or enable to refine the conclusions, would definitely increase the value and timeliness of the work.

We thank the reviewer for this positive feedback. In the revised manuscript, we also include mobility and mortality data collected between May and August 2020. The results of the analysis with the mortality data until the end of August 2020 were extremely similar to the original ones: with the new data there was a correlation of 0.62 between the *log COVID-19 Mortality Probability* and the social distancing start time (while with the data until May, the correlation was 0.64). The results are summarized in the last section of the Results “Prolonged impact of the initial response on the COVID-19 related mortality”. Of note, we used the mobility data until May for modeling the lockdown dynamics. The trend of the mobility data was quite similar across countries in the first several months of the pandemic, in which the lockdown occurred. We opted to extract features that characterize the lockdown and thus, we fitted the model to the data until May, as was presented in the original version of the manuscript. Nevertheless, our analyses of recent mortality data clearly shows that lockdown timing had a long term effect on mortality, as reflected by the mortality data from the end of August, and thus validate the implications of our model and our initial conclusions.

The reanalyses with recent mortality data were also conducted across the US states and across the 58 countries for which Apple data were available. In the Results section, page 7, we write:

“Evidently, the presented analysis corresponds to the lockdown taken as an initial response by most countries in the first several months of the pandemic. Next, we examined whether the effect of the initial response sustained over a prolonged time period. To this end, we extracted the reported mortality rates on August 31, 2020, and normalized them by the population size (termed, Aug-20 COVID-19 Mortality Probability). A significant correlation between τ , as computed from fitting $\hat{M}(t)$ to the mobility data during the lockdown period, and the log Aug-20 COVID-19 Mortality Probability, was maintained ($r^2 = 0.62$, P value = $1e - 8$ and $r^2 = 0.34$, P value = $2e - 4$ when excluding and including Japan, respectively; Fig EV4). Still, neither

the lockdown strictness nor the lockdown duration was significantly correlated with the log Aug-20 COVID-19 Mortality Probability (Table EV3). The significant correlation sustained when the log Aug-20 COVID-19 Mortality Probability was examined across the 58 countries for which data are available, across the United States countries, and when the log Aug-20 COVID-19 Infection Probability was examined (Appendix Fig S3-S5). Altogether, these analyses imply that the initial response was critical to curb total COVID-19 related mortality and had a long term impact.”

Referee #2 (Remarks for Author):

In their manuscript, Loewenthal and co-authors introduce an elegantly simple mathematical approach for exploring the relationship between COVID-19 deaths and social distancing (measured by proxy cellphone mobility indices). This article is a nice contribution but there are a few things the authors still need to address. Overall, the conclusions of the paper that I believe the authors have shown are directly tied to Figure 4. That is, I believe that the authors have shown that time to social distancing (as measured by cellphone mobility) is related to mortality (That a great conclusion!). However, I don't buy some of the other stated conclusions in the paper.

I don't buy the authors claims about the time between infection to fatality. The authors try to make a claim about this lag based on a proxy (the offset between mobility reduction and the decline in mortality growth rate). This is too much of a stretch for me, there are just too many sources of potential confounding that make this proxy not a good measure of the lag between infection to fatality. To underscore my point, it seems ridiculous to me that there would be so much variation from country to country in this value (e.g., 38-39 days in Finland, Hungary, and Estonia) but only 8 in the United Kingdom? I would recommend authors remove this section and its claims to strengthen the paper. I don't think the paper needs this added conclusion to be impactful.

We thank the reviewer for the supportive comment for strengthening our manuscript. Following this comment, we removed this section from the manuscript.

The abstract mention that the time at which social distancing was initiated explains 62% of the variation of the number of deaths. Where are the authors pulling this statistic from, it was not clear to me in the manuscript where this came from? I am skeptical of this statistic.

We thank the reviewer for highlighting this ambiguity. We referred to the correlation between the relative social distancing start time (denoted by τ) and the mortality probability (Fig. 5). Following this comment, we corrected the relevant text in the abstract to:

“the time at which social distancing was initiated is highly correlated with the number of deaths, $r^2 = 0.64$, while the lockdown strictness or its duration are not as informative.”

We also note that the previous analysis that resulted in $r^2 = 0.62$ was corrected to $r^2 = 0.64$.

Regarding the 7.49 day delay causing a doubling in deaths: Are the authors implying this statistic (7.49 days) is related to the doubling time of the epidemic? It would seem like this based on their citation of the 7.4 day from Li et al. The 7.4 day doubling time is not a great estimate (<https://www.nytimes.com/interactive/2020/03/21/upshot/coronavirus-deaths-by-country.html>; Silverman et al. Science Translational Medicine 2020 for the US at least). Initial doubling times were closer to every 3 days. However, it seems to me that what the authors are really measuring is not the same doubling time as mentioned in Li et al as it is more a function of the effective doubling time around the time that the interventions were occurring (not the initial epidemic doubling time).

The reviewer is right. We refined the description of this estimate, and now it is presented throughout the text as the estimated time that a delay in starting social distancing would lead to doubling the total number of deaths. This measure is not quite the doubling time, as the doubling time is defined to be the time until the number of deaths or confirmed cases is doubled, and the measure inferred here represents the effect of the interventions imposed on March and their effect on the mortality rates in May or August. We address this distinction as well as the discrepancy in the literature regarding the doubling time in the revised manuscript. In page 10, we write:

“The results of our analysis show that a delay of 7.49 in initiating social distancing would lead to doubling the total expected number of deaths. Interestingly, these results resemble previous results regarding the pandemic doubling time, which is described as the time that passes until the number of deaths or confirmed cases at a given time point double (Wu et al, 2020; Kraemer et al, 2020; Li et al, 2020). Others studies estimated the doubling time to be lower than four days (Lurie et al, 2020; Muniz-Rodriguez et al, 2020; Silverman et al, 2020). Notably, all of the aforementioned studies were conducted on infection or mortality data collected during different time phases between December 2019 and the beginning of March 2020, and it is possible that

the effective doubling time diverged across countries and through time. In contrast, the results presented in this study are based on the days that preceded the social distancing which was imposed on March by most countries and the number of deaths accumulated until May.”

What does the Apple mobility data actually measure? "movement" is not directly measured by cell phones. That is, is it based on a pedometer in the phone? Is it based on the maximum diameter of the convex hull of the cell phone's position on a given day? As far as I know Apple does not make this clear and it is a crucial limitation of this data -- What is the data?

Following this comment, we provide additional information regarding the meaning of the mobility data provided by apple. In page 10, we write:

“Mobility data, $M(t)$, with one data point per day, t , were downloaded from Apple repository on August 31, 2020 (Apple, 2020). The Apple dataset reports the daily volume of directions requested from Apple maps on mobile cellphones for driving, walking, or using transit (public transportation) in a specified region. The amount of requests per day is reported as the percentage with respect to a benchmark (100%) set on January 13th, 2020.”

I applaud the authors for not just assuming exponential growth dynamics (e.g., SIR models etc...) in their manuscript. However, the authors should still describe how a depletion of susceptibles could alternatively explain some of the death dynamics observed in this manuscript. For example, the growth rate of deaths must (on average) decrease as the susceptible population is exposed to the virus. As it currently stands, it seems the authors are assuming that all changes in deaths could be attributable to social distancing without accounting for the fact that there are other processes that can dictate the dynamics of deaths. I think what the authors have done is probably ok for the time-window they study but I don't think this implicit assumption would hold later in the epidemic when the resistant population is non-trivially large, this should be explicitly discussed.

We agree with the reviewer that other factors came into effect with the progress of the pandemic. Therefore, we concentrate on the initial period, in which the lockdown had the largest impact and the percentage of immune individuals is too low to have a substantial effect. However, in the revised manuscript, we also demonstrate the long-term effect of the lockdown on recently reported mortality levels. We present this analysis in the Results section, and we discuss this issue in the Discussion (page 9):

“Notably, the analysis conducted in this study corresponds to the initial period of the COVID-19 pandemic, when a small portion of the community was immune to the disease. The mortality model suggested here, $\widehat{D}(t)$ (similar to Tátrai & Várallyay, 2020), fits the dynamics observed during this period. With the accumulating information regarding means of prevention and treatment, as well as with the growing proportion of immunity within the population, the trends have changed within and across countries (Linka et al, 2020). Therefore, we do not expect that the mortality model would fit more recent and future mortality data. However, by extracting the reported death tolls approximately four months after the lockdown period (August 31, 2020), we validated that our conclusions still hold (Fig EV4). That is, our analyses suggest that the immediate lockdown response time to the pandemic outbreak is highly correlated to the death tolls in the long run.”

Regarding the authors literature review: The relation between social distancing / intervention and deaths / cases is probably one of the most studied parts of COVID-19. In fact, to the best of my knowledge, numerous authors have looked at the relation between mobility data (as a proxy for social distancing) and cases/deaths. The authors only cite Soures et al. A brief google search brings up a bunch of relevant other citations.

We have deepened our literature survey, and now we mention different studies that relied on mobile phone usage in relation with the COVID-19 pandemic. Notably, there were several studies that used various sources of mobility data, and all have shown the association of mobility and local COVID-19 outbreaks, however, they did not model the data trends in a way that enables to extract the lockdown features.

In the Introduction section, pages 3-4, we write:

“Location data collected from mobile phone calls have previously been linked with the identification of pandemic outbreaks, e.g., the 2005 cholera outbreak in Senegal (Finger et al, 2016). With the spread usage of smartphones nowadays, location and mobility data are routinely collected by numerous service providers. Mobility data from such datasets were shown to be associated with COVID-19 hotspots of disease transmission and spread (Kraemer et al, 2020; Badr et al, 2020; Pepe et al, 2020; Benzell et al, 2020; Bonaccorsi et al, 2020; Soures et al, 2020; Linka et al, 2020).”

In the Discussion section, page 8-9, we write:

“Mobility data collected from location identification of various smartphone applications have been previously analyzed in relation with the COVID-19 pandemic, e.g., to better understand the importance of travel restrictions on the infection rate or to construct platforms for capturing movements between provinces for decision making (Kraemer et al, 2020; Badr et al, 2020; Pepe et al, 2020; Benzell et al, 2020; Bonaccorsi et al, 2020; Soures et al, 2020; Linka et al, 2020). All of these studies proved that collecting mobility data is of high importance for controlling the trajectories of the pandemic. However, they focused on limited geographical areas and none has modelled the data through time in order to extract meaningful features of the lockdown.”

Minor Comments:

Throughout the manuscript the authors need to be a little more precise with their discussion of R² statistics. The square of the Pearson product moment (between observed and fitted) would be an R² statistic. The authors just need to tighten up the language on this point as it is confusing the way they have written it.

We thank the reviewer for pinpointing this. We corrected it throughout the text.

The authors only describe the definition of COVID-19 Mortality Probability in the methods. This is a key statistic in their paper and needs to be made clear in the results.

Fixed. We now clarify this in the Results section (page 6):

“To compute the expected mortality rate across time, we fitted a logistic function, denoted as $\widehat{D}(t)$, to the accumulated number of COVID-19 deaths of each country across time, $D(t)$:

$$\widehat{D}(t) = \frac{L_a}{1 + e^{-k_a(t-t_a^0)}}$$

as in Tatrai and Várallyay (2020). The parameters L_a , k_a , and t_a^0 are similar to those defined for the mobility model and represent the total expected mortality at the end of the pandemic, the mortality increase rate, and the time the cumulative mortality has reached its midpoint, respectively. This enabled to compute the COVID-19 Mortality Probability, namely, the expected mortality normalized by the population size of each country. The fitting of $\widehat{D}(t)$ to $D(t)$ across countries resulted in an average r^2 of 0.99 (max P value = $1e-96$; Appendix Table S3; see Figure 3 for examples of Israel and Japan and Appendix Figure S2 for all countries).”

On page 5, the authors must specify that their statement that the phases reflect a certain functional form is a subjective determination. E.g., it looks logistic to us. I am fine with this as I like the functional form they picked, but still, it is subjective and that must be mentioned. The current wording makes it seem objective.

Following the comment raised by the reviewer, we refined our presentation of the model. In page 5, we write:

“Inspection of the mobility trends during this time period revealed four phases: (1) a stable phase of high mobility (with fluctuations on weekends); (2) a sharp drop (suggesting social distancing has actually started); (3) a period of low mobility; and (4) a gradual incline towards a normal routine (Fig 1). Phases (1)-(3) resemble a (mirrored) logistic function and phase (4) is approximately linear. We modeled this overall trend by assembling a logistic function and a linear one as a function of time (t , given in days)”

Page 6, the authors should show the functional form of the logistic function for COVID-19 mortality. "See Methods" is fine if used sparingly but the authors do it too much and it makes the paper hard to read.

Fixed. The logistic function of the mortality model is now described in the Results section. in page 6 we write:

“To compute the expected mortality rate across time, we fitted a logistic function, denoted as $\widehat{D}(t)$, to the accumulated number of COVID-19 deaths of each country across time, $D(t)$:

$$\widehat{D}(t) = \frac{L_d}{1 + e^{-k_d(t-t_d^0)}}$$

as in Tátrai and Várallyay (2020). The parameters L_d , k_d , and t_d^0 are similar to those defined for the mobility model and represent the total expected mortality at the end of the pandemic, the mortality increase rate, and the time the cumulative mortality has reached its midpoint, respectively. This enabled to compute the COVID-19 Mortality Probability, namely, the expected mortality normalized by the population size of each country. The fitting of $\widehat{D}(t)$ to $D(t)$ across countries resulted in an average r^2 of 0.99 (max P value = $1e - 96$; Appendix Table S3; see Figure 3 for examples of Israel and Japan and Appendix Figure S2 for all countries).”

Page 9, is σ the variance of the estimate? This notation is not defined.

Fixed. The notation σ referred to the standard deviation and is now replaced with “std” throughout the text.

The new results the authors introduce in the discussion (last few paragraphs) should be in the results. The discussion should not introduce new data but summaries the data.

Done. The relevant paragraph, that presents the analysis over the 58 countries, the US states, and the infection data, was moved to the “Association between mobility and mortality data” section in the results.

The authors should denote t' and t' in Figure 1 as it is confusing with the concomitant notation for t_0 and t_1 .

Fixed. The annotations were incorporated in Figure 1.

Page 13, The results of changing the threshold of deaths (from 10 to 5 or 20) should include a reference to the relevant figures in the supplement.

Following this comment, we repeated the analysis for increasing thresholds (from one to 50) and we present the results in Fig. EV5 (see below). The graph shows that the results quite stabilize when reaching 10 deaths as a threshold. Note that the number of examined countries decreases with the use of larger thresholds, as not all countries reported more than 10 deaths during this time period:

Figure EV5. The correlation between τ and the log *COVID-19 Mortality Probability* for increasing number of deaths as thresholds. τ was computed as the difference between the day in which increasing number of deaths (x axis) were reported. For each threshold, the correlation was computed between τ and the log *COVID-19 Mortality Probability* across countries, while including or excluding Japan. The correlation was computed over the OECD countries that have sufficient data for each threshold. Iceland, Latvia, New Zealand, and Slovakia reported 10, 19, 22, and 27 deaths overall until May 10, therefore the correlation was computed for 37 countries from one to 10 deaths, 36 countries from 11 to 18 deaths, 35 countries from 19 to 21 deaths, 34 countries from 22 to 26 deaths, and for 33 countries from 27 to 50 deaths.

16th Sep 2020

Dear Dr. Avram,

Thank you for the submission of your revised manuscript to EMBO Molecular Medicine. We have now received the enclosed report from the referee who was asked to re-assess it. As you will see the reviewer is now supportive and I am pleased to inform you that we will be able to accept your manuscript pending the following final editoria amendments:

***** Reviewer's comments *****

Referee #2 (Comments on Novelty/Model System for Author):

No change in this respect from my prior review. The authors have clarified key language and I think their conclusions now stand on solid ground.

Referee #2 (Remarks for Author):

The authors have done a good job revising their manuscript. I have only a minor comment:

Regarding the 7.49 [day] doubling time. 2 points:

(1) there is is a typo in this sentence:

"The results of our analysis show that a delay of 7.49 in initiating social distancing would lead to doubling the total expected number of deaths."
The authors need to speictfy 7.49 "days".

(2) I am still unclear what this 7.49 [days] represents. Are they saying this is some type of effect size lating how the doubling time is changed with vs. without lockdown? The authors need to try again to clarify / tighten their prose here.

The authors have made all the requested editorial changes.

23rd Sep 2020

Dear Prof. Pupko,

We are pleased to inform you that your manuscript is accepted for publication and is now being sent to our publisher to be included in the next available issue of EMBO Molecular Medicine.

Corresponding Author Name: Itay Mayrose, Tal Pupko

Manuscript Number: EMM-2020-13171-V2